# Dietary Protein Levels Modulate the Antioxidant Capacity during Different Growth Stages in Huanjiang Mini-Pigs

**DOI:** 10.3390/antiox12010148

**Published:** 2023-01-07

**Authors:** Yating Liu, Md. Abul Kalam Azad, Xichen Zhao, Qian Zhu, Xiangfeng Kong

**Affiliations:** 1Key Laboratory of Agro-Ecological Processes in Subtropical Region, Hunan Provincial Key Laboratory of Animal Nutritional Physiology and Metabolic Process, Institute of Subtropical Agriculture, Chinese Academy of Sciences, Changsha 410125, China; 2College of Advanced Agricultural Sciences, University of Chinese Academy of Sciences, Beijing 100008, China; 3Research Center of Mini-Pig, Huanjiang Observation and Research Station for Karst Ecosystems, Chinese Academy of Sciences, Huanjiang 547100, China

**Keywords:** antioxidant capacity, crude protein, Huanjiang mini-pigs, plasma, small intestine

## Abstract

Adequate crude protein (CP) levels in diets play potential roles in swine production. This study determined the impacts of different CP levels on the antioxidant capacity of pigs during different body weight (BW) stages. Three hundred and sixty Huanjiang mini-pigs were allocated to one of three independent experiments, including a 5–10 kg BW group, where CP levels included 14%, 16%, 18%, 20%, and 22%; a 10–20 kg BW group, where CP levels included 12%, 14%, 16%, 18%, and 20%; and a 20–30 kg BW group, where CP levels included 10%, 12%, 14%, 16%, and 18%. These independent experiments were conducted for 28, 28, and 26 days, respectively. Results showed that the 20% CP level increased (*p* < 0.05) the plasma CAT and GSH-Px activities and the GSH concentration of pigs than in the pigs supplemented with the 14–18% CP levels, and the 20% CP level up-regulated (*p* < 0.05) the ileal oxidative stress-related gene expression levels of pigs than in the pigs supplemented with the 14% CP level at the 5–10 kg BW. In addition, diets supplemented with 18% CP level increased (*p* < 0.05) the ileal GSH concentration of pigs than in the pigs supplemented with the 14% and 20% CP levels, and the 16–18% CP levels increased (*p* < 0.05) the jejunal SOD activity of pigs than in the pigs supplemented with the 14% CP level. At 10–20 kg BW, the 16% CP level presented the strongest jejunal and ileal antioxidant capacity, the 18% CP level had the lowest plasma concentrations of MDA and highest GSH, and the 14–16% CP levels increased the plasma CAT and SOD activities (*p* < 0.05). Moreover, the 16–20% CP levels up-regulated (*p* < 0.05) the oxidative stress-related gene expression levels. At 20–30 kg BW, diets supplemented with the 16% CP level increased the plasma CAT activity of pigs than in the pigs supplemented with the 12–14% CP levels, and the 14–16% CP levels decreased the MDA concentration compared with the 10% CP levels (*p* < 0.05). In conclusion, these findings indicate adequate CP levels of 20%, 16%, and 14% for Huanjiang mini-pigs at the 5–10, 10–20, and 20–30 kg BW stages, respectively.

## 1. Introduction

Oxidative stress is described as the phenomenon of the imbalance between the production of reactive oxygen/nitrogen species (ROS/RNS) and the organism’s ability to eliminate them by the anti-oxidative defense system [1]. This stress results in a rising occurrence of free radicals and/or reduced activity of antioxidant defenses against free radicals [2]. The mammalian intestine is known as the central site for the digestion and absorption of nutrients (including proteins). A large quantity of free radicals can be produced with the absorption and metabolism of nutrients, which have an adverse impact on the intestinal mucosa [3,4]. The endogenous antioxidant enzyme system of the body, such as superoxide dismutase (SOD), glutathione peroxidase (GSH-Px), and catalase (CAT), can reduce the ROS existed in the cell liquid phase [5]. Moreover, oxidative stress is also highly correlated with the growth and development of animals during pre- and post-natal life, which influence the productivity of livestock industry.

Protein is known as a pivotal and the most important nutrient for the growth and development of mammals and is one of the prominent energy sources for livestock production. In recent years, nutritional strategies have emerged and it has been well demonstrated that they can increase the level of production and improve the health of animals and products obtained from them, as well as to enhance productivity in livestock [6,7]. Growing research evidence suggested that supplementation of low-protein diets with free amino acids showed various biological functions besides use for protein synthesis. For example, supplementing crystalline amino acids could modulate gene expression, decrease excessive body fat, improve intestinal growth, contribute to immunological defense, and minimize gut disorders [8,9]. Diets with higher protein levels could lead to excess intake of essential amino acids and nitrogen excretion in feces and urine, resulting in reduced nitrogen efficiency. In addition, the fermentation of protein in the hindgut impairs intestinal health [8]. However, inadequate crude protein (CP) levels largely affect the pig’s intestinal health. Previous studies demonstrated that mothers fed with low-protein diets caused oxidative stress in their offspring [10]. However, studies focusing on the impacts of CP levels on antioxidant capacity in pigs are lacking.

Huanjiang mini-pig, a famous Chinese indigenous pig breed, is genetically stable, resistant to diseases, and has excellent meat quality [11]. However, the ineffective growth, conventional free-range farming, and lower comprehensive rearing efficiency have greatly restricted the extensive production of Huanjiang mini-pigs. Previously we determined that adequate CP levels in diets improve the growth of Huanjiang mini-pigs during different ages [12], and higher CP levels may increase intestinal inflammatory status by activating the *TLR-MyD88-NF-κB* signaling pathway [13]. However, the impacts of different levels of dietary CP on the antioxidant capacity of Huanjiang mini-pigs still remained unknown. It has also been found that ROS and antioxidants influence the immune system, and oxidative stress could increase inflammation by generating the Th1 or Th2 cytokines [14]. Thus, we hypothesized that adequate levels of CP inclusion in diets could optimize the antioxidant capacity of pigs. Therefore, this study was carried out to evaluate the impacts of optimal dietary CP levels on the antioxidant capacity of Huanjiang mini-pigs at different growth stages, which would provide a potential approach to enhance the intestinal health of Chinese indigenous pig breeds. In addition, due to the physiological similarities of pigs to humans’, Chinese indigenous pigs are a suitable pig breed for medical models. The outcomes of this study can provide clues for the prevention and control of metabolic diseases in humans.

## 2. Materials and Methods

### 2.1. Animals, Diets, and Experimental Management

The animal experiments for this study were carried out at the Huanjiang Observation and Research Station for Karst Ecosystems, Huanjiang, Guangxi, China. A total of 360 healthy Huanjiang mini-pigs (female:male, 1:1) with similar body weight (BW) were allocated for three independent experiments, which were conducted for 28, 28, and 26 days, respectively. The three experiments were conducted at the 5–10 kg, 10–20 kg, and 20–30 kg BW stages, and there was a 5-day adaptation period for each experiment. At the 5–10 kg BW (experiment 1), 220 pigs (28 days old) with an initial BW of 5.32 ± 0.46 kg were allocated randomly into five dietary CP treatment groups. Each treatment contained 8–10 experimental pens, and each pen (3.0 × 4.5 m) consisted of five pigs. At 5–10 kg BW, pigs were fed with CP levels of 14%, 16%, 18%, 20%, and 22%. At 10–20 kg BW (experiment 2), 84 pigs (60 days old) with an initial BW of 11.27 ± 1.43 kg were allocated randomly into five dietary CP treatment groups. Each treatment group contained 15–19 experimental pens, and each pen (1.5 × 0.6 m) consisted of one pig. At the 10–20 kg BW, pigs were fed with CP levels of 12%, 14%, 16%, 18%, and 20%. At 20–30 kg BW (experiment 3), 56 pigs (94 days old) with an initial BW of 18.80 ± 2.21 kg were allocated randomly into five dietary CP treatment groups. Each treatment group contained 15–19 experimental pens, and each pen (1.5 × 0.6 m) consisted of one pig. At 20–30 kg BW, pigs were fed with CP levels of 10%, 12%, 14%, 16%, and 18%.

During the trials, pigs were fed at 08:00, 14:00, and 20:30 daily. All animals were housed in a controlled temperature (23–25 °C) and humidity (60 ± 5%) room and had *ad libitum* access to water and food at all times. All experimental animals were in good health conditions, and had no gastrointestinal diseases or any antibiotic exposure prior to the study. The feed formulation was designed for this study (Appendix A) followed by the National Research Council (NRC)-2012 [15] and the Chinese nutrient requirements of swine in China (NY/T65-2004) [16]. According to the NRC-2012 [15] recommendations, the premix formulation uses the isoenergetic of the individual components.

### 2.2. Sample Collection

At the end of each independent experiment, after 12 h from the last feeding, one pig per pen (5–10 kg BW) or group (10–20 and 20–30 kg BW) was selected based on the same average BW of the respective group. The pigs were then euthanized using commercial electrical stunning (120 V, 200 Hz; Electric pig stunner, Xinhai Electronic Technology Co., Ltd., Xinhai, China) for sampling. To obtain the plasma, 10 mL blood samples were drawn from the anterior vena cava of each pig into heparin-treated tubes (Saihua, China) and then centrifuged at 3500× *g* at 4 °C for 10 min. Afterward, the plasma samples were stored at −20 °C immediately for antioxidant indicators analysis. Approximately 2 cm of the jejunum (10 cm below the duodenum-jejunum junction) and ileum (10 cm above the ileocecal junction) tissues were collected and washed with cold physiological saline. Then the mucosa scrapings were performed, immediately transferred into frozen liquid nitrogen, and kept at −80 °C to analyze antioxidant indicators and related gene expressions.

### 2.3. Plasma and Intestinal Mucosa Oxidation-Antioxidant Indices Analysis

The plasma and small intestinal oxidation-antioxidant indices of pigs were evaluated following the previous study by Wang et al. [17]. Briefly, the jejunum and ileum mucosa samples (0.1000 g) were added to physiological saline (1 mg:9 mL) and homogenized, and then the supernatant was collected using centrifugation for 10 min at 2000× *g* and 4 °C. The activities of CAT, SOD, and GSH-Px and the concentrations of glutathione (GSH) and malondialdehyde (MDA) in plasma and small intestinal mucosa were measured by the colorimetric method with porcine-special ELISA kits (Meimian, Yancheng, China; CAT kit, the intra- and the inter-assay coefficients of variation <10 and <9%, respectively; SOD kit, the intra- and the inter-assay coefficients of variation <8 and <7%, respectively; GSH-Px kit, the intra- and the inter-assay coefficients of variation <9 and <8%, respectively; GSH kit, the intra- and the inter-assay coefficients of variation <10 and <9%, respectively; MDA kit, the intra- and the inter-assay coefficients of variation <10 and <9%, respectively). The total protein concentration of the small intestinal mucosa was determined by the Pierce BCA Protein Assay Kit (Beyotime, Shanghai, China) using the Multiscan Spectrum Spectrophotometer (Tecan, Infinite M200 Pro, Basel, Switzerland). The measured parameters of the intestinal mucosa samples were normalized to the total protein concentration of each sample.

### 2.4. RNA Extraction and Gene Expression Analysis of Small Intestinal Mucosa

The total RNA was isolated from jejunal and ileal mucosa using the TRIZOL Regent (Magen, Guangzhou, China), following the operation manual and procedure described previously [17]. After evaluation of RNA quality and quantity, a PrimeScript RT reagent kit with gDNA Eraser (TaKaRa, Dalian, China) was used to synthesize the cDNA, then the real-time PCR analysis was performed on the LightCycler^®^ 480 II Real-Time PCR System (Roche, Basel, Switzerland) [18]. The RT-PCR conditions were considered according to the previous study [19]. The pig-specific primers are shown in Appendix A, which were synthesized by Sangon Biotech (Shanghai) Co., Ltd., China. The gene expression level was evaluated using the 2^−ΔΔCt^ method [20] and using the β-actin as an internal control.

### 2.5. Data Analysis

The normal distribution of the experimental data was confirmed by the Shapiro-Wilk test before assessing differences between different CP groups. All experimental data for this study were statistically performed by one-way ANOVA, and the comparison analysis among different groups was carried out using Tukey post-hoc test with SPSS 22.0 software (SPSS, Inc., Chicago, IL, USA). The data are presented with means and their standard error (SE). The *p*-values < 0.05 were indicated as statistically significant, and 0.05 ≤ *p* < 0.10 was defined as a trend toward differences. The images were processed using the GraphPad Prism 8.0 (San Diego, CA, USA) software.

## 3. Results

### 3.1. Impacts of Dietary CP Levels on Plasma Oxidation-Antioxidant Indices of Pigs

The impacts of dietary CP levels on the plasma oxidation-antioxidant indices are shown in Table 1. At 5–10 kg BW, diets supplemented with 14% and 18% CP levels reduced the plasma CAT activity, whereas diets supplemented with 14–18% CP levels reduced the plasma GSH-Px activity of pigs compared with the 20% CP level (*p* < 0.05). Diets supplemented with 14% CP level decreased (*p* < 0.05) the plasma GSH concentration of pigs compared with the 18–22% CP levels. At 10–20 kg BW, diets supplemented with 14–18% CP levels increased (*p* < 0.05) the plasma CAT activity of pigs compared with the 20% CP level. Diets supplemented with 18% CP level increased the concentration of plasma GSH, whereas decreased plasma MDA than the other four CP levels (*p* < 0.05). Furthermore, diets supplemented with 14% and 16% CP levels increased (*p* < 0.05) the plasma SOD activity of pigs compared with the 20% CP level. At 20–30 kg BW, diets supplemented with 12% and 14% CP levels decreased the plasma CAT activity of pigs compared with the 16% CP level, and the 18% CP level reduced the plasma GSH concentration of pigs compared with the pigs of other four CP levels (*p* < 0.05).

### 3.2. Impacts of Dietary CP Levels on Jejunal Oxidation-Antioxidant Indices of Pigs

The impacts of different dietary CP levels on the jejunal oxidation-antioxidant indices are listed in Table 2. At 5–10 kg BW, diets supplemented with 14% CP level reduced (*p* < 0.05) the jejunal SOD activity of pigs compared with the 16–20% CP levels. Dietary supplementation with 14% and 20% CP levels reduced (*p* < 0.05) the concentration of jejunal GSH of pigs compared with the pigs supplemented with 16–18% CP levels. At 10–20 kg BW, diets supplemented with 14–18% CP levels increased the jejunal SOD activity, 16–18% CP levels increased CAT activity, and 14–20% CP levels increased GSH-Px activity of pigs, when compared with the 12% CP level (*p* < 0.05). Diets supplemented with 14% and 20% CP levels reduced (*p* < 0.05) the jejunal GSH concentration of pigs compared with the 16% CP level. At 20–30 kg BW, diets supplemented with 10% CP level increased (*p* < 0.05) the jejunal MDA concentration of pigs compared with the pigs supplemented with 14–16% CP levels, while other antioxidant indexes were not significantly changed (*p* > 0.05) with different levels of dietary CP supplementation.

### 3.3. Impacts of Dietary CP Levels on Ileal Oxidation-Antioxidant Indices of Pigs

The impacts of different dietary CP levels on the ileal oxidation-antioxidant indices are shown in Table 3. At 5–10 kg BW, diets supplemented with 14%, 20%, and 22% CP levels reduced the ileal GSH concentration of pigs compared with the 18% CP level, whereas 14% and 20% CP levels had higher ileal GSH concentration compared with the 22% CP level (*p* < 0.05). At 10–20 kg BW, diets supplemented with 14% and 20% CP levels reduced the CAT activity, whereas 12%, 14%, and 20% CP levels reduced the SOD activity of pigs compared with the 16–18% CP levels (*p* < 0.05). Moreover, diets supplemented with 16% CP level showed a higher ileal GSH-Px activity of pigs compared with the other four CP levels, while 20–22% CP levels reduced MDA concentration in the pigs compared with the 14% CP level (*p* < 0.05). At 20–30 kg BW, diets supplemented with 10% and 18% CP levels reduced the ileal SOD activity of pigs compared with the 12–16% CP levels, whereas 10% CP level increased MDA concentration of pigs compared with the 14–16% CP levels (*p* < 0.05). Moreover, diets supplemented with the 12% CP level increased (*p* < 0.05) the ileal MDA concentration of pigs compared with the 14% CP level.

### 3.4. Impacts of Dietary CP Levels on Jejunal Oxidative Stress-Related Gene Expression of Pigs

The impacts of different dietary CP levels on the gene expressions related to oxidative stress in jejunal mucosa are shown in Figure 1 and Figure 2. At 5–10 kg BW, dietary 14–16% CP levels down-regulated (*p* < 0.05) the *Nrf1* expression, whereas 14% and 20% CP levels down-regulated (*p* < 0.05) the *Nrf2* expression in the jejunum of pigs compared with the pigs supplemented with the 22% CP level (Figure 2A). At 10–20 kg BW, dietary 12% CP level displayed a trend for down-regulation of the jejunal *CuZnSOD* (*p* = 0.088) and *GPX-1* (*p* = 0.077) expressions compared with the 20% CP level (Figure 1B). The 12–14% CP levels down-regulated (*p* < 0.05) the *Nrf1* expression in the jejunum compared with the 20% CP level, whereas the 12% CP level down-regulated (*p* < 0.05) the *CAT* expression in the jejunum compared with the 18–20% CP levels and the jejunal *Nrf2* expression compared with the 16–20% CP levels (Figure 1B and Figure 2B). Moreover, diets supplemented with the 12% CP level down-regulated (*p* < 0.05) the *NQO1* expression in the jejunum compared with the 16% CP level (Figure 2B). At 20–30 kg BW, diets supplemented with 12% and 18% CP levels down-regulated (*p* < 0.05) the jejunal *CAT* expression of pigs compared with the 16% CP level (Figure 1C).

### 3.5. Impacts of Dietary CP Levels on Ileal Oxidative Stress-Related Gene Expression of Pigs

The impacts of different dietary CP levels on the gene expressions related to oxidative stress in the ileal mucosa of pigs are presented in Figure 3 and Figure 4. At 5–10 kg BW (Figure 3A and Figure 4A), diets supplemented with 14–18% CP levels down-regulated the *CAT* expression, and the 14% CP level down-regulated the *GPX-1* expression in the ileal mucosa of pigs (*p* < 0.05), when compared with the 20–22% CP levels (*p* < 0.05). Diets supplemented with 14% CP level down-regulated the *CuZnSOD* expression, and 14–18% CP levels down-regulated the *GPX-4* expression in the ileum of pigs compared with the 22% CP level, whereas diets consisting of 14% CP level down-regulated the *MnSOD2* expression compared with the 20% CP level (*p* < 0.05). In addition, 14–16% CP levels down-regulated the *GPX-4* expression in the ileum of pigs compared with the 20–22% CP levels (*p* < 0.05). The 14–18% CP levels down-regulated the ileal *Nrf1* and *Nrf2* expressions compared with the 20% CP level, while the 14% CP level down-regulated the *Nrf1* expression in the ileum compared with the 16% CP level (*p* < 0.05). Furthermore, diets supplemented with 20% CP level down-regulated the expression of *KEAP1* in the ileum compared with the 14% CP level, while up-regulated the expression of *NQO1* in the ileum compared with the other four CP levels (*p* < 0.05). At 10–20 kg BW (Figure 3B and Figure 4B), diets supplemented with 12% CP level down-regulated the ileal *GPX-1* expression compared with the 14%, 18%, and 20% CP levels, and the 12–14% CP levels down-regulated the ileal *Nrf1* expression compared with the 20% CP level (*p* < 0.05). At 20–30 kg BW (Figure 3C and Figure 4C), diets supplemented with 10% CP level down-regulated (*p* < 0.05) the ileal *Nrf1* expression of pigs compared with the pigs supplemented with 14% CP level.

## 4. Discussion

Oxidative stress is one of the most critical circumstances in livestock production. In commercial production, pigs are generally exposed to weaning, nutritional, and environmental stressors, which could induce oxidative stress [5], while diet can alter endogenous antioxidant capacity by modulating the ability to ROS [21]. Therefore, the present study analyzed the oxidation-antioxidant balance status in plasma and small intestinal mucosa to investigate the impacts of different levels of dietary CP on the antioxidant capacity during different growth stages in Huanjiang mini-pigs. Our findings revealed that the positive effects of adequate dietary CP level on the antioxidant capacity are partly attributed to the up-regulated gene expression levels of antioxidant enzymes, which might also be associated with the *Nrf2* and *KEAP1* signaling molecules.

The antioxidant system consists of antioxidant enzymes (e.g., SOD, GSH-Px, and CAT) and non-enzymatic antioxidants (GSH) [22]. The SOD is known as the first line of defense against excessive oxidative radicals and could catalyze the transformation of superoxide radicals into H_2_O_2_, which is transformed into H_2_O and O_2_ by GSH-Px and CAT [23]. A recent study reported that dietary protein sources could influence antioxidant capacity by increasing SOD activity, due to their different amino acid composition and bioactive substances [24]. In the present study, the 14% CP diet reduced the plasma CAT and GSH-Px activities and GSH concentration compared with the 20% CP diet at the 5–10 kg BW, indicating that the diet containing 14% CP decreased antioxidant capacity by depressing non-enzymatic and enzymatic antioxidant defense components. In addition, 16% and 18% CP diets improved the antioxidant capacity of pigs by increasing the jejunal SOD activity and GSH concentration of pigs in comparison to pigs supplemented with 20% CP level, and the 18% CP diet improved antioxidant capacity by increasing the jejunal GSH concentration compared to the other four CP levels. These results were consistent with a previous study [25], which found that weaning stress decreased the serum GSH-Px and SOD activities of piglets. Moreover, Zhang and Piao [24] also reported that a lower CP level in diets significantly decreased the serum CuZnSOD, MnSOD, CAT, and GSH-Px activities and GSH concentration of piglets. However, Wu et al. [25] found that dietary CP levels did not affect the serum CAT, MDA, and T-AOC levels in pigs. The reason may be that SOD converts superoxide radicals into H_2_O_2_, and then GSH converts H_2_O_2_ into water with GSH-Px. However, the 14% CP diet lacked sufficient protein to synthesize antioxidant enzymes, for example, SOD and GSH-Px [9]. Inadequate protein nutrition can decrease the production of antioxidant enzymes, while weaned piglets are sensitive to a diet with low CP levels.

The endogenous oxidative damage is reflected by blood antioxidant capacity, and the higher antioxidant capacity has beneficial effects on alleviating oxidative stress [8,26]. In the present study, pigs fed with 12% and 20% CP diets had the lowest plasma antioxidant capacity, whereas 20% CP diet decreased the plasma SOD and CAT activities of pigs compared with the 16% and 18% CP levels at the 10–20 kg BW. In addition, the 12% CP diet decreased small intestinal mucosa antioxidant capacity by decreasing the jejunal SOD, CAT, and GSH-Px activities and GSH concentration, as well as the ileal SOD and GSH-Px activities. Moreover, the 14% and 20% CP diets impaired the ileal antioxidant capacity, while the 16% and 18% CP diets improved both jejunal and ileal antioxidant capacity compared to the other CP levels. These results suggest that protein deficiency could lead to a reduction in the antioxidant enzyme synthesis capacity and affect the antioxidant capacity of pigs, while excessive dietary protein could also aggravate the metabolic load, produce more free radicals, and aggravate oxidative stress of pigs. Optimal dietary CP levels provided favorable conditions to synthesize antioxidant enzymes and promoted GSH synthesis by increasing the GSH rate-limiting enzyme (such as GCL) biosynthesis [27].

At 20–30 kg BW, the 16% CP diet increased the plasma CAT activity of pigs compared with the 12% and 14% CP diets, while the 18% CP diet decreased the plasma GSH concentration of pigs compared with the other four CP levels. These findings indicate that diets supplemented with 16% CP level increased while 18% CP level decreased the plasma antioxidant capacity of pigs. In addition, the 12–16% CP diets improved the ileal antioxidant capacity of pigs by increasing the SOD activity compared to the pigs supplemented with 10% and 18% CP levels. The decreased antioxidant capacity due to higher dietary CP levels maybe result from the disturbance of the pig’s nitrogen metabolism and production of more free radicals. At the same time, inadequate CP levels could also affect the production of antioxidant enzymes.

MDA, a biomarker of oxidative stress, is the main oxidation product of ROS and its level directly reflects the degree of oxidative damage [9]. In the present study, the 18% CP diet decreased the plasma MDA concentration of pigs than in the pigs supplemented with other four CP levels, as well as the ileal MDA concentration compared to the 12% CP diet at the 10–20 kg BW. Moreover, the 14% and 16% CP diets improved the jejunal and ileal antioxidant capacity of pigs by decreasing the MDA concentration compared to the 10% CP diet at the 20–30 kg BW. These findings suggest that optimal dietary CP levels could alleviate the pig’s oxidative damage by decreasing the MDA concentration in the body. A previous study also found that high/low CP levels increased growing quail’s plasma MDA concentration [28].

Previous studies indicated that the expression levels of genes related to antioxidant capacity are tightly related to the antioxidant enzyme activities [29]. Therefore, to further explore the molecular mechanism of dietary CP levels affecting the gene expressions related to antioxidant capacity in the small intestinal mucosa of Huanjiang mini-pigs, we investigated the gene expression of *KEAP1-Nrf2* signaling pathways. The *Nrf2* is considered an oxidative stress protector and is a key transcription factor that maintains the gene expression of antioxidant enzymes [30]. The *KEAP1* has been known as a cytosolic binding protein for *Nrf2* [31], which could promote the ubiquitination-proteasomal degradation of *Nrf2* [32]. In the present study, the 22% CP diet up-regulated the jejunal *Nrf1* and *Nrf2* expressions, and the 20% and 22% CP diets up-regulated the ileal *Nrf2*-regulated genes involved in the antioxidant defense system, including *CAT*, *NQO1*, *MnSOD2*, *CuZnSOD2*, *GPX-1*, *GPX-4*, *Nrf1*, and *Nrf2* expressions at the 5–10 kg BW. At 10–20 kg BW, the 12% CP diet down-regulated several antioxidant-related genes in the jejunum, while the 16%, 18%, and 20% CP diets up-regulated those genes. Moreover, the 16% CP diet improved the jejunal antioxidant capacity by up-regulating the *CAT* expression, and the 14% CP diet improved the ileal antioxidant capacity by up-regulating the *Nrf1* expression at the 20–30 kg BW. Taken together, these findings suggest that optimal dietary CP levels, on the one hand, may improve the antioxidant ability of pigs by up-regulating *Nrf2* expression to up-regulate antioxidant enzyme genes; on the other hand, may enhance the antioxidant enzyme activities by up-regulating the related gene expression levels. It has been reported that the *KEAP1* can negatively regulate *Nrf2* expression [30], which was also observed at the 5–10 kg BW in the present study. Moreover, the gene expressions related to intestinal antioxidant capacity were not consistent with the intestinal mucosal antioxidant indices evaluated in the present study. However, further studies are needed to evaluate those absolute proteins in order to correlate with the antioxidant capacity indices and are also necessary to clarify their mechanism of action.

## 5. Conclusions

According to the results gathered in the current study, both lower and higher crude protein levels could negatively affect the antioxidant capacity during different growth stages in Huanjiang mini-pigs. The positive effects of adequate CP levels on the antioxidant capacity are partly attributed by up-regulating the gene expression levels of antioxidant enzymes, some of which may be regulated through several signaling molecules, such as *Nrf2* and *KEAP1*. However, more in-depth research is necessary to reveal the underlying mechanisms of the impacts of CP levels on Huanjiang mini-pigs. Finally, these findings indicate that the optimal dietary CP levels during the 5–10, 10–20, and 20–30 kg BW stages were 20, 16, and 14%, respectively, for Huanjiang mini-pigs.

## Figures and Tables

**Figure 1 antioxidants-12-00148-f001:**
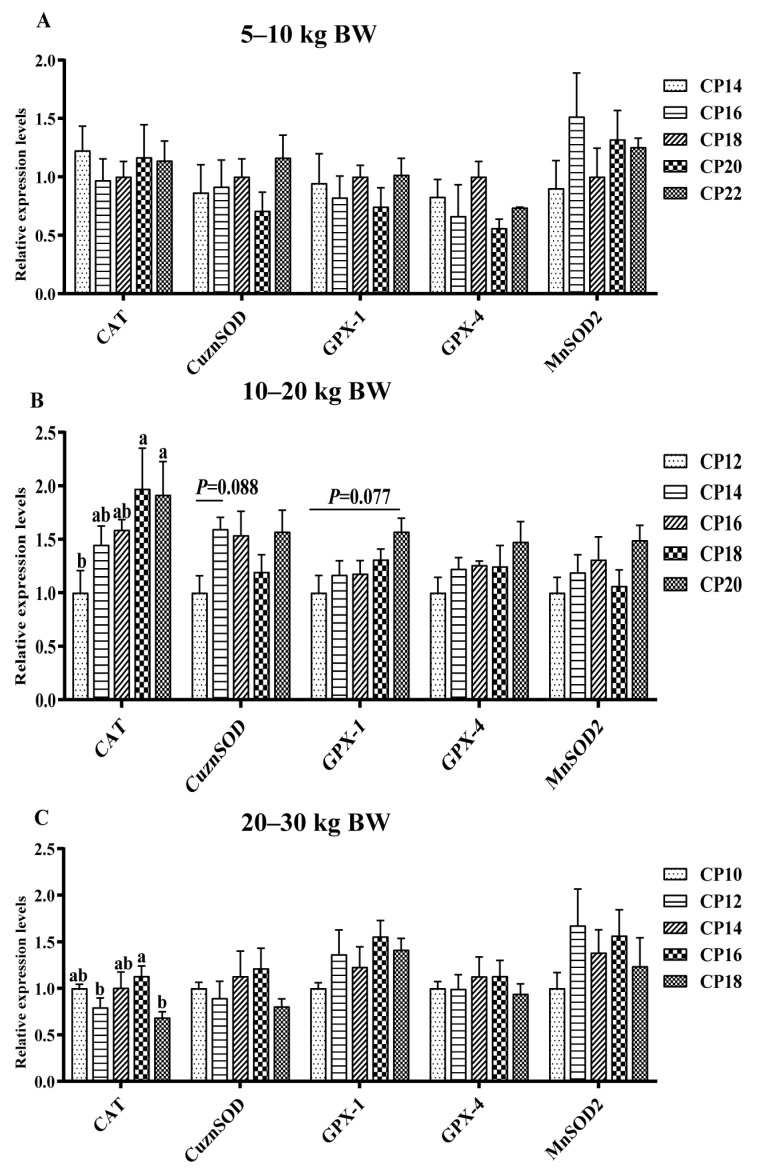
Impacts of dietary crude protein (CP) levels on the oxidative stress-related gene expression in the jejunal mucosa of Huanjiang mini-pigs during the 5–10 kg (**A**), 10–20 kg (**B**), and 20–30 kg (**C**) BW stages. Data are represented as means ± SE (*n* = 6–8 per group). Different superscript letters indicate significant differences among the five groups (*p* < 0.05). *CAT*, catalase; *CuznSOD*, copper- and zinc-containing superoxide dismutase; *GPX-1*, glutathione peroxidase 1; *GPX-4*, glutathione peroxidase 4; *MnSOD*, manganese-containing superoxide dismutase.

**Figure 2 antioxidants-12-00148-f002:**
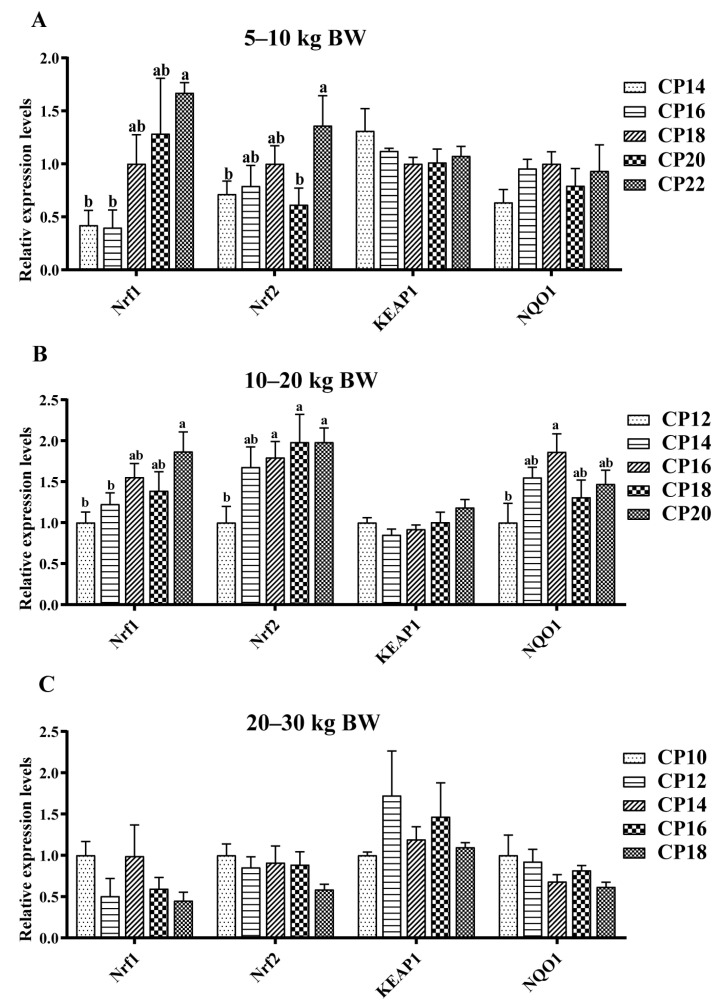
Impacts of dietary crude protein (CP) levels on the Nrf1/2-KEAP1-NQO1-related gene expression in jejunal mucosa of Huanjiang mini-pigs during the 5–10 kg (**A**), 10–20 kg (**B**), and 20–30 kg (**C**) BW stages. Data are represented as means ± SE (*n* = 6–8 per group). Different superscript letters indicate significant differences among the five groups (*p* < 0.05). *Nrf1*, nuclear factor erythroid 2-related factor 1; *Nrf2*, nuclear factor erythroid 2-related factor 2; *KEAP1*, Kelchlike ECH associated protein 1; *NQO1*, nicotinamide adenine dinucleotide (phosphate).

**Figure 3 antioxidants-12-00148-f003:**
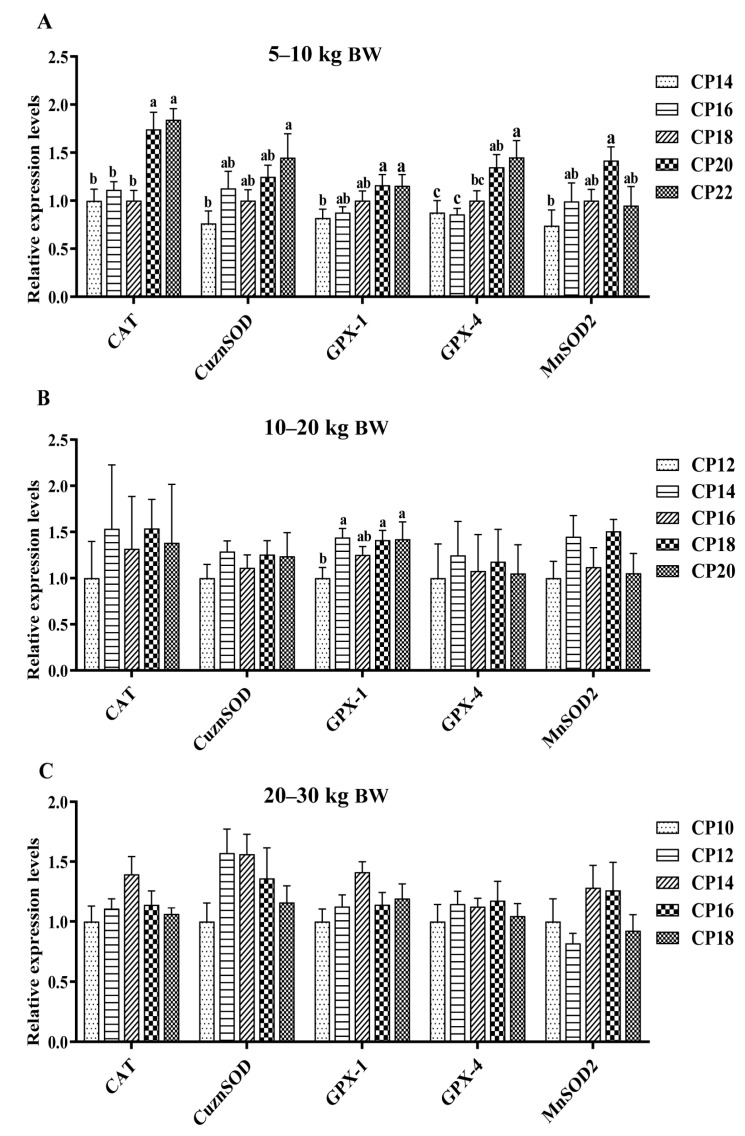
Impacts of dietary crude protein (CP) levels on the oxidative stress-related gene expressions in the ileal mucosa of Huanjiang mini-pigs during the 5–10 kg (**A**), 10–20 kg (**B**), and 20–30 kg (**C**) BW stages. Data are represented as means ± SE (*n* = 6–8 per group). Different superscript letters indicate significant differences among the five groups (*p* < 0.05). *CAT*, catalase; *CuZnSOD*, copper- and zinc-containing superoxide dismutase; *GPX-1*, glutathione peroxidase 1; *GPX-4*, glutathione peroxidase 4; *MnSOD*, manganese-containing superoxide dismutase.

**Figure 4 antioxidants-12-00148-f004:**
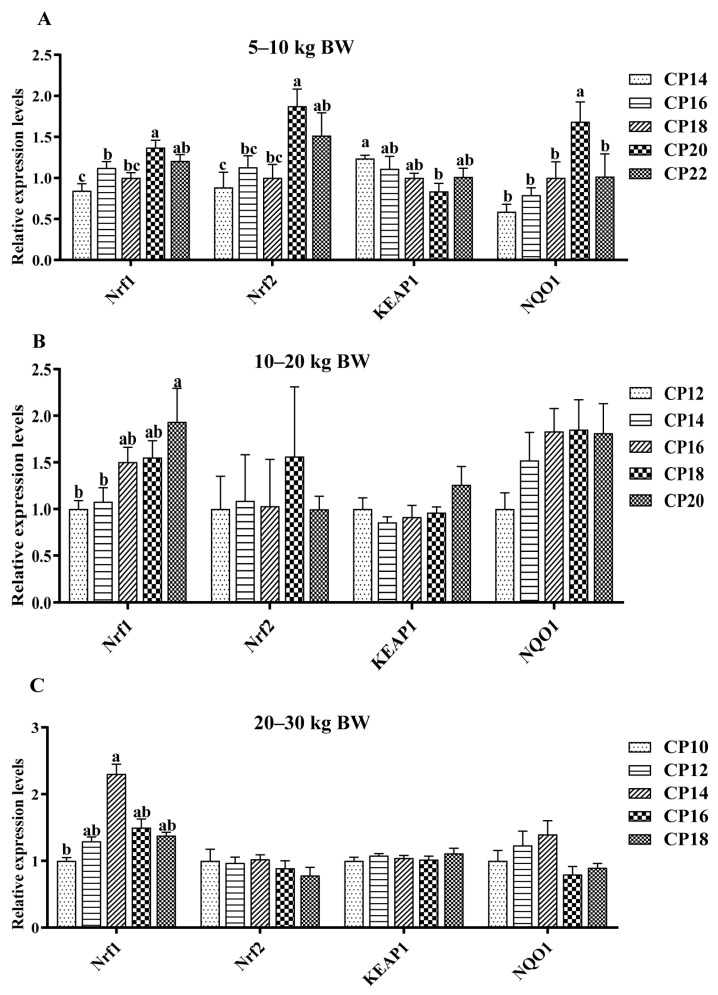
Impacts of dietary crude protein (CP) levels on the Nrf1/2-KEAP1-NQO1-related gene expressions in the ileal mucosa of Huanjiang mini-pigs during the 5–10 kg (**A**), 10–20 kg (**B**), and 20–30 kg (**C**) BW stages. Data are represented as means ± SE (*n* = 6–8 per group). Different superscript letters indicate significant differences among the five groups (*p* < 0.05). *Nrf1*, nuclear factor erythroid 2-related factor 1; *Nrf2*, nuclear factor erythroid-2 related factor 2; *KEAP1*, Kelch-like ECH associated protein 1; *NQO1*, nicotinamide adenine dinucleotide (phosphate).

**Table 1 antioxidants-12-00148-t001:** Impacts of dietary crude protein (CP) levels on the plasma oxidation-antioxidant indices of Huanjiang mini-pigs during different growth stages.

Items	Dietary CP Levels (%)					*p*-Values
5–10 kg BW	14	16	18	20	22	
CAT (U/mL)	6.44 ± 0.32 ^b^	7.89 ± 0.68 ^ab^	7.02 ± 0.53 ^b^	8.98 ± 0.44 ^a^	7.67 ± 0.68 ^ab^	0.026
GSH (μg/mL)	58.80 ± 1.90 ^b^	75.13 ± 6.60 ^ab^	86.47 ± 8.57 ^a^	98.28 ± 12.66 ^a^	94.44 ± 8.49 ^a^	0.020
GSH-Px (U/mL)	78.94 ± 4.32 ^b^	85.39 ± 4.28 ^b^	83.88 ± 5.76 ^b^	100.39 ± 3.63 ^a^	92.30 ± 3.38 ^ab^	0.049
MDA (nmol/mL)	8.84 ± 0.45	9.17 ± 0.28	9.14 ± 0.31	9.07 ± 0.37	9.88 ± 0.38	0.400
SOD (U/mL)	108.78 ± 9.08	96.92 ± 7.48	102.57 ± 7.64	108.23 ± 7.56	113.09 ± 7.68	0.679
10–20 kg BW	12	14	16	18	20	
CAT (U/mL)	7.90 ± 0.52 ^ab^	9.41 ± 0.44 ^a^	9.15 ± 0.68 ^a^	9.52 ± 0.66 ^a^	7.36 ± 0.45 ^b^	0.030
GSH (μg/mL)	95.92 ± 8.18 ^b^	90.48 ± 6.65 ^b^	91.97 ± 5.71 ^b^	117.97 ± 5.42 ^a^	79.57 ± 4.82 ^b^	0.002
GSH-Px (U/mL)	94.94 ± 9.13	96.56 ± 3.78	96.38 ± 7.78	111.94 ± 6.07	90.98 ± 6.31	0.312
MDA (nmol/mL)	9.63 ± 0.44 ^a^	9.62 ± 0.62 ^a^	9.45 ± 0.27 ^a^	8.03 ± 0.47 ^b^	10.34 ± 0.38 ^a^	0.029
SOD (U/mL)	99.28 ± 7.85 ^ab^	121.42 ± 8.76 ^a^	124.34 ± 4.95 ^a^	118.88 ± 8.20 ^ab^	95.18 ± 11.18 ^b^	0.049
20–30 kg BW	10	12	14	16	18	
CAT (U/mL)	9.57 ± 0.84 ^ab^	9.02 ± 0.34 ^b^	7.88 ± 0.41 ^b^	10.90 ± 0.67 ^a^	9.66 ± 0.67 ^ab^	0.020
GSH (μg/mL)	104.66 ± 4.54 ^a^	101.71 ± 7.66 ^a^	111.60 ± 7.14 ^a^	115.14 ± 2.20 ^a^	78.14 ± 3.30 ^b^	0.004
GSH-Px (U/mL)	113.45 ± 10.00	115.74 ± 6.98	105.39 ± 9.59	115.31 ± 7.63	110.39 ± 7.46	0.709
MDA (nmol/mL)	8.12 ± 0.67	7.94 ± 0.56	7.67 ± 0.54	8.31 ± 0.59	8.68 ± 0.34	0.801
SOD (U/mL)	149.56 ± 10.52	151.91 ± 6.33	146.50 ± 7.88	141.69 ± 8.26	135.80 ± 14.32	0.805

Data are presented as means ± SE (*n* = 6–8 per group). Different superscript letters indicate significant differences among the five groups (*p* < 0.05). BW, body weight; CAT, catalase; GSH, glutathione; GSH-Px, GSH peroxidase; MDA, malondialdehyde; SOD, superoxide dismutase.

**Table 2 antioxidants-12-00148-t002:** Impacts of dietary crude protein (CP) levels on the jejunal mucosa oxidation-antioxidant indices of Huanjiang mini-pigs during different growth stages.

Items	Dietary CP Levels (%)					*p*-Values
5–10 kg BW	14	16	18	20	22	
CAT (U/mL)	3.92 ± 0.20	4.33 ± 0.27	4.44 ± 0.23	4.41 ± 0.30	4.13 ± 0.30	0.606
GSH (μg/mL)	29.22 ± 2.52 ^b^	40.95 ± 1.78 ^a^	40.53 ± 2.52 ^a^	31.81 ± 2.15 ^b^	33.93 ± 2.30 ^ab^	0.008
GSH-Px (U/mL)	43.22 ± 3.50	53.91 ± 2.64	44.83 ± 2.49	51.68 ± 2.30	46.08 ± 1.50	0.058
MDA (nmol/mL)	3.67 ± 0.16	3.20 ± 0.22	3.91 ± 0.30	3.43 ± 0.24	3.35 ± 0.20	0.257
SOD (U/mL)	19.45 ± 1.83 ^b^	32.16 ± 2.27 ^a^	31.69 ± 1.53 ^a^	30.81 ± 2.46 ^a^	25.78 ± 2.87 ^ab^	0.004
10–20 kg BW	12	14	16	18	20	
CAT (U/mL)	4.02 ± 0.16 ^b^	4.69 ± 0.36 ^ab^	5.18 ± 0.28 ^a^	5.04 ± 0.29 ^a^	4.75 ± 0.36 ^ab^	0.045
GSH (μg/mL)	35.13 ± 1.35 ^b^	40.71 ± 2.51 ^ab^	45.80 ± 1.71 ^a^	39.68 ± 1.98 ^ab^	34.07 ± 2.43 ^b^	0.011
GSH-Px (U/mL)	41.86 ± 2.94 ^b^	57.57 ± 3.44 ^a^	56.23 ± 3.65 ^a^	57.74 ± 2.30 ^a^	53.22 ± 2.34 ^a^	0.004
MDA (nmol/mL)	3.21 ± 0.34	3.18 ± 0.07	2.92 ± 0.25	3.15 ± 0.26	3.47 ± 0.23	0.633
SOD (U/mL)	23.37 ± 2.87 ^b^	35.54 ± 4.85 ^a^	34.79 ± 1.92 ^a^	37.76 ± 2.71^a^	30.10 ± 3.01 ^ab^	0.045
20–30 kg BW	10	12	14	16	18	
CAT (U/mL)	4.62 ± 0.30	4.76 ± 0.27	4.65 ± 0.29	4.71 ± 0.21	4.73 ± 0.51	0.997
GSH (μg/mL)	47.77 ± 3.40	40.78 ± 2.26	50.56 ± 0.71	42.04 ± 3.68	42.56 ± 3.42	0.102
GSH-Px (U/mL)	58.05 ± 3.78	56.40 ± 3.99	56.27 ± 4.56	55.25 ± 2.93	63.19 ± 3.28	0.647
MDA (nmol/mL)	3.24 ± 0.23 ^a^	2.82 ± 0.24 ^ab^	2.31 ± 0.16 ^b^	2.27 ± 0.23 ^b^	2.60 ± 0.21 ^ab^	0.023
SOD (U/mL)	41.65 ± 4.99	37.29 ± 1.99	42.91 ± 2.25	41.20 ± 1.88	45.52 ± 4.31	0.538

Data are presented as means ± SE (*n* = 6–8 per group). Different superscript letters indicate significant differences among the five groups (*p* < 0.05). BW, body weight; CAT, catalase; GSH, glutathione; GSH-Px, GSH peroxidase; MDA, malondialdehyde; SOD, superoxide dismutase.

**Table 3 antioxidants-12-00148-t003:** Impacts of dietary crude protein (CP) levels on the ileal mucosa oxidation-antioxidant indices of Huanjiang mini-pigs during different growth stages.

Items	Dietary CP Levels (%)					*p*-Values
5–10 kg BW	14	16	18	20	22	
CAT (U/mL)	3.46 ± 0.20	4.06 ± 0.26	4.03 ± 0.34	3.97 ± 0.32	4.07 ± 0.22	0.539
GSH (μg/mL)	37.10 ± 2.41 ^b^	40.58 ± 0.82 ^ab^	44.12 ± 1.85 ^a^	36.08 ± 3.29 ^b^	29.66 ± 1.26 ^c^	0.001
GSH-Px (U/mL)	40.87 ± 2.70	38.87 ± 2.35	41.98 ± 2.41	45.06 ± 3.57	40.20 ± 2.21	0.513
MDA (nmol/mL)	3.27 ± 0.28	2.94 ± 0.15	2.72 ± 0.12	2.63 ± 0.13	2.86 ± 0.19	0.177
SOD (U/mL)	29.66 ± 2.00	29.96 ± 1.76	25.27 ± 2.96	31.97 ± 3.99	22.79 ± 2.34	0.157
10–20 kg BW	12	14	16	18	20	
CAT (U/mL)	4.69 ± 0.41 ^ab^	4.12 ± 0.22 ^b^	4.97 ± 0.22 ^a^	5.07 ± 0.15 ^a^	4.13 ± 0.43 ^b^	0.049
GSH (μg/mL)	35.30 ± 3.53	38.56 ± 2.97	48.79 ± 3.06	42.94 ± 4.74	42.47 ± 4.79	0.219
GSH-Px (U/mL)	48.66 ± 2.99 ^b^	45.61 ± 2.64 ^b^	60.49 ± 1.95 ^a^	47.19 ± 3.79 ^b^	50.00 ± 4.55 ^b^	0.049
MDA (nmol/mL)	3.12 ± 0.16 ^a^	2.78 ± 0.17 ^ab^	2.88 ± 0.21 ^ab^	2.44 ± 0.21 ^b^	2.43 ± 0.12 ^b^	0.045
SOD (U/mL)	24.54 ± 1.85 ^b^	27.14 ± 3.53 ^b^	41.08 ± 2.76 ^a^	37.85 ± 3.96 ^a^	27.67 ± 3.64 ^b^	0.004
20–30 kg BW	10	12	14	16	18	
CAT (U/mL)	4.22 ± 0.18	4.48 ± 0.25	4.88 ± 0.37	4.71 ± 0.17	4.43 ± 0.30	0.433
GSH (μg/mL)	43.56 ± 3.23	43.66 ± 3.61	46.76 ± 2.79	41.74 ± 3.36	38.54 ± 4.79	0.638
GSH-Px (U/mL)	55.22 ± 3.30	53.97 ± 2.57	53.73 ± 3.01	55.50 ± 3.51	47.28 ± 2.78	0.316
MDA (nmol/mL)	2.56 ± 0.17 ^a^	2.47 ± 0.08 ^ab^	1.89 ± 0.22 ^c^	1.99 ± 0.19 ^bc^	2.30 ± 0.21 ^abc^	0.049
SOD (U/mL)	28.89 ± 3.23 ^b^	46.53 ± 1.15 ^a^	47.25 ± 5.36 ^a^	49.19 ± 2.59 ^a^	32.55 ± 2.22 ^b^	0.001

Data are presented as means ± SE (*n* = 6–8 per group). Different superscript letters indicate significant differences among the five groups (*p* < 0.05). BW, body weight; CAT, catalase; GSH, glutathione; GSH-Px, GSH peroxidase; MDA, malondialdehyde; SOD, superoxide dismutase.

## Data Availability

The data presented in this study are included in the article. Further inquiries can be directed to the corresponding authors.

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
