# Peer review of "Dietary Protein Levels Modulate the Antioxidant Capacity during Different Growth Stages in Huanjiang Mini-Pigs"

_antioxidants, 2023, doi:10.3390/antiox12010148_

Round 1

Reviewer 1 Report

In this study, the authors evaluated the impacts of dietary crude protein (CP) levels on the antioxidant capacity of Huanjiang mini-pigs during different growth stages. The authors found that adequate dietary CP levels can partly attribute to the antioxidant capacity of pigs by modulating the gene expression levels of antioxidant enzymes. Moreover, their findings also indicate that the CP levels of 20%, 16%, and 14% were the optimal levels for the 5-10 kg, 10-20 kg, and 20-30 kg BW Huanjiang mini-pigs. It’s a very interesting study; however, the manuscript contains some concerns that need to be addressed before considering publication.

In this study, the authors mainly focused on antioxidant related indicators. Are these indicators related to animal growth performance? This point should be discussed.

In the experiment, pigs at different growth stages (different weight) were used, but  it the age information is also important.

Please show the P values in the tables or figures.

L 33-35: not necessary for abstract. I recommend to delete this part from the abstract.

L 85-94: Provide the dimensions of pens for each trial; as the number of pigs per pen was different.

L 95-100: Include more detailed rearing conditions for pigs; for example, room temperature, humidity, ventilation, etc.

L 105:  Name and manufacturing company of that electrical stunning equipment?

L 105-106: Blood samples were drawn from where? Provide the specific location.

L 111: The term “expression” should be “expressions”.

L 121: Please cross-check the company, city, and country names.

L 122: Add city name.

L 139/142: Keep consistent format; United States/USA.

L 150-151: This statement is not correct with the results; cross-check with the results.

L 188-191: The 20% CP level also reduced the GSH concentration.

L 195-196: 18-20% CP level reduced MDA concentration compared with which group?

Section 3.4-3.5: Please indicate specific Figure numbers to make them more clear to the reader. For example, Figure 1A, B, C, etc.

Figure legends should be revised as: 5-10 kg growth stage >> 5-10 kg BW

Author Response

Response to the Reviewer’s comments 1

In this study, the authors evaluated the impacts of dietary crude protein (CP) levels on the antioxidant capacity of Huanjiang mini-pigs during different growth stages. The authors found that adequate dietary CP levels can partly attribute to the antioxidant capacity of pigs by modulating the gene expression levels of antioxidant enzymes. Moreover, their findings also indicate that the CP levels of 20%, 16%, and 14% were the optimal levels for the 5-10 kg, 10-20 kg, and 20-30 kg BW Huanjiang mini-pigs. It’s a very interesting study; however, the manuscript contains some concerns that need to be addressed before considering publication.

Response: We would like to thank you for your valuable and insightful comments and suggestions on our manuscript. The comments and suggestions are helpful to improve the quality of our manuscript. We have read all comments carefully and made revisions accordingly. Please be noted that all of the changes are indicated track changes.

In this study, the authors mainly focused on antioxidant related indicators. Are these indicators related to animal growth performance? This point should be discussed.

Response: Thanks for your valuable suggestions. We added more details of indicators related to animal growth performance in the Introduction part of the revised manuscript (L 49-51).

In the experiment, pigs at different growth stages (different weight) were used, but  it the age information is also important.

Response: We added the piglet's age with different growth stages (L 93-103). The piglets were 28, 60, and 94 days old at the 5-10, 10-20, and 20-30 kg BW stages, respectively.

Please show the P values in the tables or figures.

Response: We added P values in the Tables and Figures.

L 33-35: not necessary for abstract. I recommend to delete this part from the abstract.

Response: We deleted this part from the abstract.

L 85-94: Provide the dimensions of pens for each trial; as the number of pigs per pen was different.

Response: We added the dimensions of pens for each trial (L 95-105).

L 95-100: Include more detailed rearing conditions for pigs; for example, room temperature, humidity, ventilation, etc.

Response: We added more details in the revised manuscript (L 106-110).

L 106-110: “All animals were housed in a controlled temperature (23°C-25°C) and humidity (60±5%) room and had ad libitum access to water and food at all times. All experimental animals were in good health conditions, and had no gastrointestinal diseases or any antibiotic exposure prior to the study”.

L 105:  Name and manufacturing company of that electrical stunning equipment?

Response: We added the name and manufacturing company of the electrical stunning equipment (Electric pig stunner, Xinhai Electronic Technology Co., Ltd., Xinhai, China) L 119-120.

L 105-106: Blood samples were drawn from where? Provide the specific location.

Response: Blood samples were drawn from the anterior vena cava of each pig into heparin-treated tubes. We added this information in the revised manuscript (L 120-122).

L 111: The term “expression” should be “expressions”.

Response: We changed the term “expression” to “expressions” L 128.

L 121: Please cross-check the company, city, and country names.

Response: We are sorry for this mistake. We corrected it carefully (L 138).

L 122: Add city name.

Response: We added the city name (L 143).

L 139/142: Keep consistent format; United States/USA.

Response: Thanks for your clarifications. We checked throughout the text carefully.

L 150-151: This statement is not correct with the results; cross-check with the results.

Response: Thanks for your clarifications. We have corrected the statement (L 173-174).

L 188-191: The 20% CP level also reduced the GSH concentration.

Response: We have corrected in the revised manuscript (L 212-215).

L 195-196: 18-20% CP level reduced MDA concentration compared with which group?

Response: Thanks for your clarifications. We have corrected in the text (L 217-220).

Section 3.4-3.5: Please indicate specific Figure numbers to make them more clear to the reader. For example, Figure 1A, B, C, etc.

Response: We added the specific figure numbers in the text to make them more clear to the reader.

Figure legends should be revised as: 5-10 kg growth stage >> 5-10 kg BW

Response: We have modified the figure legends according to the reviewer’s suggestions.

Reviewer 2 Report

In the study entitled “Dietary Protein Levels Modulate the Antioxidant Capacity during Different Growth Stages in Huanjiang Mini-Pigs” the Authors’ purpose was to investigate the impacts of different crude protein levels on the antioxidant capacity of three hundred and sixty Huanjiang mini-pigs at different body weight stages.

According to the findings obtained in the survey, Authors concluded that the positive impacts of an adequate crude protein diet on the antioxidant capacity are partly attributed by enhancing the gene expression levels of antioxidant enzymes, which might be associated with Nrf2 and KEAP1 signaling molecules.

I think that the subject of the work is of interest and that the topic of the manuscript is appropriate for the Journal. The information is of significant interest to the Journal's readers. However, some changes could improve the manuscript. Therefore, I suggest that the study could be suitable for publication after minor revision.

Specific comments

The title as well as keywords accurately reflects the major findings of the work.

The abstract adequately summarize methodology, results, and significance of the study. However, Authors should indicate statistical analysis applied on the data. Authors should indicate the P values as well.

The introduction section falls within the topic of the study, however, Authors should enhance this section adding more information concerning the breeding strategy studies focused on the improvement of animal reproduction and production  and on the diet supplementation in veterinary field emphasizing the significant increase of interest showed by scientific community on diet improvement to enhance animal health status and welfare. On this regard, after the sentence Lines 49-50 “Protein is known as a pivotal and most important nutrient for the growth and development of mammals and is one of the prominent energy sources for livestock production.” I suggest to add “In recent years, nutritional strategies have emerged and it has been well demonstrated that they can increase the level of production and improve the health of animals and products obtained from them as well as to enhance productivity in livestock (D’Alessandro E. et al., Frontiers in Veterinary Science 9, 2022,1046101; Giannetto C. et al., Antioxidants 11, 2022, 2339).”

The section of Materials and Methods is clear for the reader however, some clarifications are needed and some missing information should be added.

Did Authors evaluate the health status of enrolled animals? Please clarify this aspect in the method section.

Lines 105-107: Please improve the sentence “To obtain the plasma, 10 mL blood samples were drawn from each pig and then centrifuged at 3,500 xg g at 4 °C for 10 min.” by better specifying blood collection (i.e. how blood sampling was performed (from jugular vein?), tubes used for blood collection…)

Line 119: please indicate the intra- and inter-assay variability for ELISA tests.

Regarding statistical analysis, did Authors test the conformity to normal distribution of data by a Normality test? Please clarify this aspect and specify whether data passed normality test indicating the P value.

Results section as well as Discussion section is clear and well written. The findings obtained in the study were well discussed and justified with appropriate references.

The conclusion section is well written, indeed, Authors well summarize the results and the significance of the study. Please changes  “In summary” with “According to the results gathered in the current study, ”

The tables are generally good and well represent the results of the study, whereas, Authors should improve significantly the quality of figures.

Authors should check and standardize the references in the list according to journal guidelines.

Author Response

Response to the Reviewer’s comments 2

In the study entitled “Dietary Protein Levels Modulate the Antioxidant Capacity during Different Growth Stages in Huanjiang Mini-Pigs” the Authors’ purpose was to investigate the impacts of different crude protein levels on the antioxidant capacity of three hundred and sixty Huanjiang mini-pigs at different body weight stages.

According to the findings obtained in the survey, Authors concluded that the positive impacts of an adequate crude protein diet on the antioxidant capacity are partly attributed by enhancing the gene expression levels of antioxidant enzymes, which might be associated with Nrf2 and KEAP1 signaling molecules.

I think that the subject of the work is of interest and that the topic of the manuscript is appropriate for the Journal. The information is of significant interest to the Journal's readers. However, some changes could improve the manuscript. Therefore, I suggest that the study could be suitable for publication after minor revision.

Response: Sincerest thanks for your valuable time and constructive comments on our manuscript. We have read all comments and suggestions carefully and made corrections accordingly. Please be noted that all the changes in the revised manuscript and indicated track change.

Specific comments

The title as well as keywords accurately reflects the major findings of the work.

Response: Thanks for your positive feedback.

The abstract adequately summarize methodology, results, and significance of the study. However, Authors should indicate statistical analysis applied on the data. Authors should indicate the P values as well.

Response: Thanks for your constructive suggestions. We added statistical analysis applied on the data and presented P values in the abstract.

The introduction section falls within the topic of the study, however, Authors should enhance this section adding more information concerning the breeding strategy studies focused on the improvement of animal reproduction and production  and on the diet supplementation in veterinary field emphasizing the significant increase of interest showed by scientific community on diet improvement to enhance animal health status and welfare. On this regard, after the sentence Lines 49-50 “Protein is known as a pivotal and most important nutrient for the growth and development of mammals and is one of the prominent energy sources for livestock production.” I suggest to add “In recent years, nutritional strategies have emerged and it has been well demonstrated that they can increase the level of production and improve the health of animals and products obtained from them as well as to enhance productivity in livestock (D’Alessandro E. et al., Frontiers in Veterinary Science 9, 2022,1046101; Giannetto C. et al., Antioxidants 11, 2022, 2339).”

Response: We thank the reviewer for insightful suggestions. We have further improved the Introduction part focused on the improvement of animal reproduction and production on diet supplementation (L 54-56).

The section of Materials and Methods is clear for the reader however, some clarifications are needed and some missing information should be added.

Response: We added more details in the Materials and Methods in the revised version of the manuscript.

Did Authors evaluate the health status of enrolled animals? Please clarify this aspect in the method section.

Response: All experimental pigs were in good health conditions, and had no gastrointestinal diseases or any antibiotic exposure prior to the study. We added this information in the method section (L 107-110).

Lines 105-107: Please improve the sentence “To obtain the plasma, 10 mL blood samples were drawn from each pig and then centrifuged at 3,500 xg g at 4 °C for 10 min.” by better specifying blood collection (i.e. how blood sampling was performed (from jugular vein?), tubes used for blood collection…)

Response: Thanks for your clarifications. We added more details in the revised manuscript.

L 120-122: To obtain the plasma, 10 mL blood samples were drawn from the anterior vena cava of each pig into heparin-treated tubes (Saihua, China) and then centrifuged at 3,500× g at 4 °C for 10 min.

Line 119: please indicate the intra- and inter-assay variability for ELISA tests.

Response: Thanks for your valuable suggestions. We indicated the intra- and the inter-assay coefficients of variation for all ELISA kits in the revised manuscript.

L 134-141: “The activities of CAT, SOD, and GSH-Px and the concentrations of glutathione (GSH) and malondialdehyde (MDA) in plasma and small intestinal mucosa were measured by the colorimetric method with porcine-special ELISA kits (Meimian, Yancheng, China; CAT kit, the intra- and the inter-assay coefficients of variation <10 and <9%, respectively; SOD kit, the intra- and the inter-assay coefficients of variation <8 and <7%, respectively; GSH-Px kit, the intra- and the inter-assay coefficients of variation <9 and <8%, respectively; GSH kit, the intra- and the inter-assay coefficients of variation <10 and <9%, respectively; MDA kit, the intra- and the inter-assay coefficients of variation <10 and <9%, respectively).”

Regarding statistical analysis, did Authors test the conformity to normal distribution of data by a Normality test? Please clarify this aspect and specify whether data passed normality test indicating the P value.

Response: Regarding statistical analysis, we tested the normality distribution of data by a Normality test. We have further clarified this issue in the revised manuscript (L 159-160).

Results section as well as Discussion section is clear and well written. The findings obtained in the study were well discussed and justified with appropriate references.

Response: Thanks for your positive feedback.

The conclusion section is well written, indeed, Authors well summarize the results and the significance of the study. Please changes  “In summary” with “According to the results gathered in the current study, ”

Response: Thanks for your insightful suggestions. We changed “In summary” to “According to the results gathered in the current study” L 408.

The tables are generally good and well represent the results of the study, whereas, Authors should improve significantly the quality of figures.

Response: Thanks for your valuable suggestions. We further improved the quality of the figures.

 Authors should check and standardize the references in the list according to journal guidelines.

Response: We checked the reference format carefully.

Reviewer 3 Report

This study aims to determine the impacts of different CP levels on the antioxidant capacity of pigs during different body weight (BW) stages. As the results of their experiments, dietary CP levels affected some of indices of antioxidant activities in each of growing stages of the mini pigs. Nevertheless, the authors need to explain some background information for introduction. They also need to discuss more about the results, as described below:

1. Why did the CP levels differently affect antioxidant indices and/pr gene expression? Please discuss this especially in gene network aspect.

2. Comparing the results in Tables and Figures (1 & 3), there are some discrepancy between the results of antioxidant indices and gene expression. Please explain the differences in the results between gene expression and indices.

3. Authors frequently cite precious study of grass carp. To evaluate appropriately, the present results should be compared with these animals such as pigs, chicken, and cattle, rather than carps. Do authors consider this appropriate? 

Author Response

Response to the Reviewer’s comments 3

This study aims to determine the impacts of different CP levels on the antioxidant capacity of pigs during different body weight (BW) stages. As the results of their experiments, dietary CP levels affected some of indices of antioxidant activities in each of growing stages of the mini pigs. Nevertheless, the authors need to explain some background information for introduction. They also need to discuss more about the results, as described below:

Response: Sincerest thanks for your valuable time and constructive comments on our manuscript. We have read all comments and suggestions carefully and made corrections accordingly. Please be noted that all the changes in the revised manuscript and indicated track change.

  1. Why did the CP levels differently affect antioxidant indices and/pr gene expression? Please discuss this especially in gene network aspect.

Response: Different CP levels differently affect the antioxidant capacity indices and/or gene expression due to their different availability of amino acids and bioactive substances. We have further discussed this in the revised manuscript (L 314-320).

  1. Comparing the results in Tables and Figures (1 & 3), there are some discrepancies between the results of antioxidant indices and gene expression. Please explain the differences in the results between gene expression and indices.

Response: Thanks for your concerns and insightful suggestions. These differences of the antioxidant indices and gene expression might be related to the processing time of those samples and also the determination methods. However, further studies are needed to evaluate of those absolute proteins in order to correlate with the antioxidant capacity indices and also necessary to clarify their mechanism of action. We have further discussed in the revised manuscript (L 399-403).

  1. Authors frequently cite precious study of grass carp. To evaluate appropriately, the present results should be compared with these animals such as pigs, chicken, and cattle, rather than carps. Do authors consider this appropriate? 

Response: Thanks for your clarifications. We have removed the statement related to the study of grass carp in the revised manuscript. Furthermore, we added relevant research work related to pigs and chickens.